# Mortality among 5 to 19-year-olds in rural Mali

Jenny X. Liu[1]*, Yacouba Samake[2], Oumar Tolo[2], Emily Treleaven[3], Belco Poudiougou[2], Caroline Whidden[2,4], Ari Johnson[2,5], Kassoum Kayentao[2,6], David C. Boettiger[1,7]

1 Institute for Health & Aging, School of Nursing, University of California, San Francisco, San Francisco, California, United States of America, 2 Muso Health, Bamako, Mali, 3 Survey Research Center, Institute for Social Research, University of Michigan, Ann Arbor, Michigan, United States of America, 4 Department of Disease Control, London School of Hygiene and Tropical Medicine, London, United Kingdom, 5 Department of Medicine, University of California, San Francisco, San Francisco, California, United States of America, 6 Malaria Research and Training Center, University Sciences, Techniques, and Technologies of Bamako, Bamako, Mali, 7 The Kirby Institute, University of New South Wales, Sydney, Australia

* Jenny.Liu2@ucsf.edu

## Abstract

The unique healthcare needs of 5 to 9-year-olds and adolescents (10–19 years) in low- and middle-income countries have been largely neglected. We generated estimates of 5 to 9-year-old and adolescent mortality in rural Mali, a setting with high under-five mortality, and aimed to define associated individual and household risk factors. We analysed cross-sectional baseline household survey data from the ProCCM trial (NCT02694055) conducted in Bankass District, Mali collected in December 2016 and January 2017. Deaths in the preceding five years, household information, and women's birth histories were documented. Factors associated with 5 to 9-year-old and adolescent mortality were analysed using Cox regression. Our study population comprised 23,485 children aged 5 to 9-years-old and 17,910 adolescents from 7,720 households. The 5 to 9-year-old and adolescent mortality rates were 3.10 and 1.90 deaths per 1,000 person-years, respectively. Mortality rates were similar among males and females aged 5 to 9 years, but grew increasingly divergent in adolescence (1.69 and 2.17 per 1,000 person-years, respectively). Five to 9-year-olds in households with untreated water had a higher risk of death than those in households with treated water. Adolescents living in the poorest households had a higher risk of death than those in the wealthiest, and adolescents in households in which no women received schooling had a higher risk of death than those in which women had some schooling. The risk of mortality was especially acute among female adolescents compared to their male counterparts, with low access to education for women being a strong contributing factor.

## Introduction

Despite global commitments to reducing the burden of disease among low- and middle-income countries (LMICs) [1], 5 to 9-year-olds and adolescents (age 10–19 years) are populations that have often been neglected by researchers and policymakers. Middle childhood and adolescence are key development periods in the life course, with health outcomes in

**Data availability statement:** Muso Inc. (https://www.musohealth.org/) own the data

used for this study and granted access to the authors. These data are not publicly available to maintain participant privacy but de-identified data will be made available to external researchers upon reasonable request. Please direct enquiries to the Director of Research at Muso Inc. (Arsene Brunelle Sandi, asandie@musohealth.org).

**Funding:** The trial was funded with resources received by Muso (OT, BP, CW, AJ) though unrestricted funding as well as dedicated research funding from Child Relief International Foundation, Grand Challenges Canada (awards 1808-17345 and TTS-2002-37264), Johnson & Johnson Foundation (awards 85442 and 82844) and USAID Development Innovation Ventures (grant number 7200AA20FA00020). Child Relief International Foundation served as the nonlegal sponsor of the trial. The funders had no role in study design, data collection and analysis, decision to publish, or preparation of the manuscript.

**Competing interests:** The authors have declared that no competing interests exist.

**Abbreviations:** CHW, community health workers; HRs, hazard ratios; LMICs, low- and middle-income countries; PHC, primary health center.

this period shaping health, educational attainment, socio-economic status, and well-being in adulthood. Renewed efforts to focus on the needs of adolescents in LMICs have spurred a recent surge in research on their mental health [2], reproductive health [3], socioeconomic wellbeing [4], and schooling [5], often in the context of child development during the transition to adulthood.

Yet, these studies constitute only a small fraction of the literature compared to other age groups, particularly children under five years of age. Very few studies estimate mortality rates in 5 to 9-year-olds and adolescence [6,7]; vital registration data can be of poor quality, particularly in LMICs where mortality rates are elevated. Using modelling approaches to address data quality challenges, an analysis of cause of death among of 5 to 9-year-olds and adolescents in LMICs suggests that injuries and non-communicable disease comprise a greater proportion of cause of death than for children under five [8]. Correspondingly, these age groups are absent in many population health surveys in LMICs. Demographic Health Surveys, for example, include older female adolescents (age 15–19) in reference to fertility, reproductive, and maternal health, and nothing about male adolescents. UNICEF's Multiple Indicators Cluster Survey has included education, development, labour, and disciplining for those aged 5–17 years, and more recently added questions about functional health [9]. A limited number of targeted and ad hoc population-level surveys in LMICs have detailed measures of physical or mental health for middle childhood and adolescents [10], including the 2021 Global Youth Survey [11]. Yet, recent analyses show that communicable diseases continue to drive mortality into middle childhood while mental health and physical injuries become increasingly relevant in adolescence [8]. Little information exists to understand the overall health profiles of 5 to 9-year-olds and adolescents in many settings globally, and particularly in many of the poorest countries, though a growing body of studies have called attention to an array of multi-level risk and protective factors [12–14].

The current analysis forms part of a larger study evaluating the effect of proactive case detection by community health workers (CHW) on under-five mortality in the Bankass health district [15]. Bankass is situated in the Mopti region of Mali, an area that relies heavily on agriculture and that serves as an important crossroad between the country's north, south, and bordering countries [16]. It is one of the poorest regions of Mali and has one of the country's highest burdens of under-five child mortality [17,18]. However, little is known about the health outcomes and needs of 5 to 9-year-olds and adolescents in this high morbidity, high mortality setting. As such, we aimed to (i) describe the trends in mortality for 5 to 9-year-olds and adolescents, and (ii) identify the associated individual and household risk factors associated with mortality in 5 to 9-year-olds and adolescents. Our findings are discussed in light of the dearth of population-level data on 5 to 9-year-olds and adolescents, and highlight areas of critical need for further research and health programming to improve survival among this group.

## Methods

### Household survey data

We analysed household census and survey data collected cross-sectionally at the baseline of a three-year cluster randomized controlled trial conducted in 137 village-clusters distributed across seven of the 22 primary health center (PHC) catchment areas in Bankass (Kanibonzon, Ende, Dimbal, Doundé, Soubala, Koulongon, and Lessagou). The study area has a population of approximately 100,000 people. The trial primarily aims to determine whether door-to-door proactive case detection by CHWs reduces under-five mortality compared to passive, site-based CHW care offered under the standard Integrated Community Case Management

protocol [19]. The trial is registered with ClincialTrials.gov (NCT02694055). Ethical approval was obtained from the Ethics Committee of the Faculty of Medicine, Pharmacy and Dentistry, University of Bamako (2016/03/CE/FMPOS). The main consideration for this analysis was the potential for data access permitting identification of study participants. The University of California, San Francisco exempted secondary analysis of the trial data from ethical approval based on the thorough deidentification of participants. All participants provided written informed consent prior to enrolling in the study, which included the baseline survey.

None of the authors had access to information that could identify individuals during or after data collection. Further details on the trial protocol are available elsewhere [15].

As a part of the trial procedures, a census was conducted among all households in the study area at baseline and updated every 12 months thereafter during the three-year study period. All households were censused between December 2016 and January 2017, just before the launch of the intervention, to enumerate all permanent residents (present more than 50% of the time in the past year). The census included a household roster to collect the age, date of birth, and sex of permanent residents, as well as information about deaths in each household in the past five years. One female member of the household was invited to complete a household survey of socio-economic factors. All women in the household aged 15–49 years (i.e., women of reproductive age) with no plans to leave the study area during the trial period who provided written informed consent were eligible to partake in the baseline survey. No children were surveyed. Surveys were adapted from the Malian Demographic and Health Surveys [18]. Women were given a reading test to assess literacy and asked if they contributed to household decision-making, if their husband had more than one wife or partner, and if they felt their husband hitting or beating them was justified under certain circumstances. Household distance to the nearest healthcare center was determined using orthodromic (great circle) distance estimates between family compound and healthcare center GPS coordinates. We supplemented the baseline survey data with information to accurately calculate the outcome from the 12-month follow-up survey, administered from February to April 2018, which used the same structure as the baseline survey but added details on women's lifetime birth histories (i.e., probes to distinguish between live and still births, clarification of multiple births, and greater precision on birth dates). Lifetime birth history refers to the number of births a woman has had in her life, the dates they occurred, and their outcome. This additional data enabled us to correct any missing data in the birth histories recorded at baseline. Survey instruments for the baseline and 12-month follow-up surveys are available for reference [17].

## Inclusion criteria

Children aged 5 to 9 years and adolescents (age 10 to 19 years) included in the analysis were those reported in lifetime birth histories by women who participated in the baseline and 12-month follow-up surveys as additional birth history data collected at 12 months (see *Household Survey Data* above) was required to accurately calculate baseline mortality rates. Based on dates of birth, all persons who were or would have been aged 5 to 19 years for ≥1 day in the five years prior to baseline were included in our analyses.

## Variable definitions

Household wealth was defined in quintiles using a principal components analysis of household possession of durable goods [20]. Women were considered to contribute to household decisions if they indicated sole or joint participation in decision-making in any of four domains—their own healthcare, visiting family or relatives, use of husband/partner's earnings, and household purchasing. Literacy was categorized based on the highest level of

reading ability among women surveyed within the household (any, partial, full). Schooling was categorized as having had any formal schooling versus none. Polygamy was defined as a survey respondent in the household indicating that her husband/partner had more than one wife/partner. Women were considered to have tolerant views towards spousal violence if they indicated that their husband hitting or beating them was justified under any of seven circumstances evaluated in the survey (going out without telling, neglecting the children, arguing, refusing sex, burning food, using contraception covertly, arguing with partners' parents).

We used the World Health Organization definitions of improved (e.g., household connections, public standpipes, boreholes, protected dug wells, protected springs and rainwater collection) and unimproved (e.g., unprotected wells or springs, surface water, vendor-or tanker truck-provided) water supply and sanitation [21]. We combined this classification of facilities with self-reported application of water treatment for safer drinking (e.g., boiling, adding sterilizing chemicals, filtering, and solar disinfection) to create four classifications: improved and treated; improved, but untreated; unimproved, but treated; unimproved and untreated. Roofing, wall, and flooring materials were defined as finished, rudimentary, or natural as per DHS definitions [22]. Distance to the nearest primary health center was defined as the Euclidean distance from the household village to the closest primary health center. All distance calculations were conducted in QGIS Version 3.4.6-Madeira (QGIS Development Team (2019), QGIS Geographic Information System, Open Source Geospatial Foundation Project, http://qgis.osgeo.org).

## Statistical analysis

The primary outcome was death. Household-level factors analyzed were ethnicity, wealth quintile, decision making contribution of women in household, highest level of reading ability among women in household, schooling among women in household, polygamy, domestic violence tolerance, water source, sanitation method, roofing material, wall material, flooring material, electricity access, primary cooking fuel, recent food shortage, livestock ownership, access to motorized transport, and distance of household from nearest healthcare center.

Cumulative probability of death from age 5 to 19 years was plotted for males and females separately using Kaplan–Meier curves. Cox regression models were used to evaluate factors associated with death. In our 5 to 9-year-old models, children were considered at risk of death from the date they turned 5-years-old or at the beginning of the 5-year period prior to baseline if aged 5 to 9 years at the time. Follow-up was censored on the day children turned 10-years-old or at the time their mother completed the baseline survey, whichever came first. In our adolescent models, adolescents were considered at risk of death from the date they turned 10-years-old or at the beginning of the 5-year period prior to baseline if aged 10 to 19 years at the time. Follow-up was censored on the day children turned 20-years-old or at the time their mother completed the baseline survey, whichever came first.

Regression models were estimated separately by age group (i.e., 5 to 9-year-olds and adolescents) and by sex (i.e., males and females). All regression analyses were adjusted for clustering at the household level using robust standard errors. Covariates were subject to bivariate analysis and those with an adjusted p-value ≤0.05 were retained in our multivariate models. Age (years) and sex were included *a priori* in all multivariate models. Five to 9-year-olds and adolescents with missing covariate data were included in analyses but hazard ratios (HRs) for missing categories are not reported. Multicollinearity was evaluated by calculating variance inflation factors for each covariate included in our final models.

We further provide summary statistics for marriage, contraception, and pregnancy indicators among female adolescents of reproductive age (15–19 years) included in the women's survey to investigate possible additional risk factors for female adolescent mortality.

Analyses were conducted with Stata 16 (Stata Corp., College Station, Texas).

## Results

### Study population characteristics

Of 15,839 households censused at baseline, 268 (1.7%) were excluded because they did not have baseline and 12-month follow-up survey data, 2,632 (16.6%) did not include a woman of reproductive age, and 5,219 (33.0%) did not include a 5 to 9-year-old or adolescent in the past five years. This left a final sample of 7,720 households, 23,485 children aged 5 to 9 years (12,060 male and 11,425 female), and 17,910 adolescents (9,823 male and 8,087 female). Table 1 presents further details of our study population.

**Table 1. Household characteristics for 5 to 9-year-olds and adolescents in Bankass.**

| | | 5 to 9-year-olds (N = 23,485) | 10 to 19-year-olds (N = 17,910) |
|---|---|---|---|
| **Ethnicity** | Dogon | 21,928 (93.4) | 16,709 (93.3) |
| | Fulani | 1,067 (4.5) | 830 (4.6) |
| | Other | 490 (2.1) | 371 (2.1) |
| **Wealth quintile** | Wealthiest | 5,396 (23.0) | 3,683 (20.6) |
| | Wealthy | 4,569 (19.5) | 2,928 (16.3) |
| | Middle | 4,509 (19.2) | 3,264 (18.2) |
| | Poor | 4,168 (17.7) | 3,498 (19.5) |
| | Poorest | 4,783 (20.4) | 4,477 (25.0) |
| | Unknown | 60 (0.3) | 60 (0.3) |
| **Decision making contribution of women in household** | Do not contribute | 15,568 (66.3) | 11,650 (65.0) |
| | Contribute | 7,114 (30.3) | 5,664 (31.6) |
| | Unknown | 803 (3.4) | 596 (3.3) |
| **Highest level of reading ability among women in household** | Can read | 918 (3.9) | 852 (4.8) |
| | Can partly read | 611 (2.6) | 489 (2.7) |
| | Cannot read | 21,059 (89.7) | 15,908 (88.8) |
| | Unknown | 897 (3.8) | 661 (3.7) |
| **Highest level of schooling among women in household** | Schooling | 2,183 (9.3) | 1,884 (10.5) |
| | No schooling | 20,497 (87.3) | 15,421 (86.1) |
| | Unknown | 805 (3.4) | 605 (3.4) |
| **Polygamy** | Monogamous | 11,189 (47.6) | 8,370 (46.7) |
| | Polygamous | 11,143 (47.4) | 8,587 (47.9) |
| | Unknown | 1,153 (4.9) | 953 (5.3) |
| **Domestic violence** | Not tolerated | 5,320 (22.7) | 4,028 (22.5) |
| | Tolerated | 17,091 (72.8) | 13,057 (72.9) |
| | Unknown | 1,074 (4.6) | 825 (4.6) |
| **Water source** | Improved and treated | 4,024 (17.1) | 3,211 (17.9) |
| | Improved but untreated | 8,858 (37.7) | 6,785 (37.9) |
| | Unimproved but treated | 1,969 (8.4) | 1,522 (8.5) |
| | Unimproved and untreated | 8,491 (36.2) | 6,276 (35.0) |
| | Unknown | 143 (0.6) | 116 (0.6) |
| **Sanitation** | Improved | 11,788 (50.2) | 9,241 (51.6) |
| | Unimproved | 11,552 (49.2) | 8,544 (47.7) |
| | Unknown | 145 (0.6) | 125 (0.7) |
| **Roofing material** | Natural | 766 (3.3) | 609 (3.4) |
| | Rudimentary | 2,101 (8.9) | 1,610 (9.0) |
| | Finished | 20,506 (87.3) | 15,578 (87.0) |
| | Unknown | 112 (0.5) | 113 (0.6) |

*(Continued)*

**Table 1.** (Continued)

| | | 5 to 9-year-olds (N = 23,485) | 10 to 19-year-olds (N = 17,910) |
|---|---|---|---|
| **Wall material** | Natural | 7,741 (33.0) | 5,868 (32.8) |
| | Rudimentary | 2,348 (10.0) | 1,777 (9.9) |
| | Finished | 13,008 (55.4) | 9,982 (55.7) |
| | Unknown | 388 (1.7) | 283 (1.6) |
| **Flooring material** | Natural | 22,289 (94.9) | 17,013 (95.0) |
| | Rudimentary | 350 (1.5) | 268 (1.5) |
| | Finished | 815 (3.5) | 600 (3.4) |
| | Unknown | 31 (0.1) | 29 (0.2) |
| **Electricity** | Yes | 8,515 (36.3) | 6,635 (37.0) |
| | No | 14,970 (63.7) | 11,275 (63.0) |
| **Primary cooking fuel** | Wood | 19,954 (85.0) | 15,144 (84.6) |
| | Straw | 3,109 (13.2) | 2,428 (13.6) |
| | Animal dung | 260 (1.1) | 191 (1.1) |
| | Other | 162 (0.7) | 147 (0.8) |
| **Food shortage in past 30 days** | Yes | 19,961 (85.0) | 15,116 (84.4) |
| | No | 3,524 (15.0) | 2,794 (15.6) |
| **Livestock** | Any | 21,947 (93.5) | 16,988 (94.9) |
| | Cows/bulls | 16,593 (70.7) | 13,189 (73.6) |
| | Horses/donkeys/mules | 17,164 (73.1) | 13,731 (76.7) |
| | Goats | 11,939 (50.8) | 9,550 (53.3) |
| | Sheep | 18,453 (78.6) | 14,574 (81.4) |
| | Chickens | 17,285 (73.6) | 13,525 (75.5) |
| **Motorized transport** | Any | 12,591 (53.6) | 9,719 (54.3) |
| | Motorbike/scooter | 12,546 (53.4) | 9,692 (54.1) |
| | Car/truck | 176 (0.7) | 136 (0.8) |
| **Nearest healthcare center, kilometers** | <2 | 4,289 (18.3) | 3,403 (19.0) |
| | 2–4.99 | 6,194 (26.4) | 4,553 (25.4) |
| | 5–6.99 | 5,592 (23.8) | 3,982 (22.2) |
| | 7–9.99 | 4,599 (19.6) | 3,619 (20.2) |
| | ≥10 | 2,811 (12.0) | 2,353 (13.1) |

All values are n (%N).

## Death by age and sex

Fig 1 shows that males and females died at a similar rate between the ages of 5 to 8 years. From 9 years of age, the probability of death among females became greater than that of similarly aged males and this gap continued to widen throughout adolescence.

## Factors associated with death in 5 to 9-year-olds

In the five years prior to baseline, a total of 180 deaths occurred among 5 to 9-year-olds at a rate of 3.10 per 1,000 person years (2.95 and 3.27 per 1,000 person years for males and females, respectively; further disaggregated rates are given in S1 and S2 Tables). Table 2 shows that 5 to 9-year-olds living in households with an untreated water source had a higher risk of death than those living in households with a treated water source (adjusted hazard ratio [aHR] 1.61 for improved but untreated versus improved and treated, 95% confidence interval

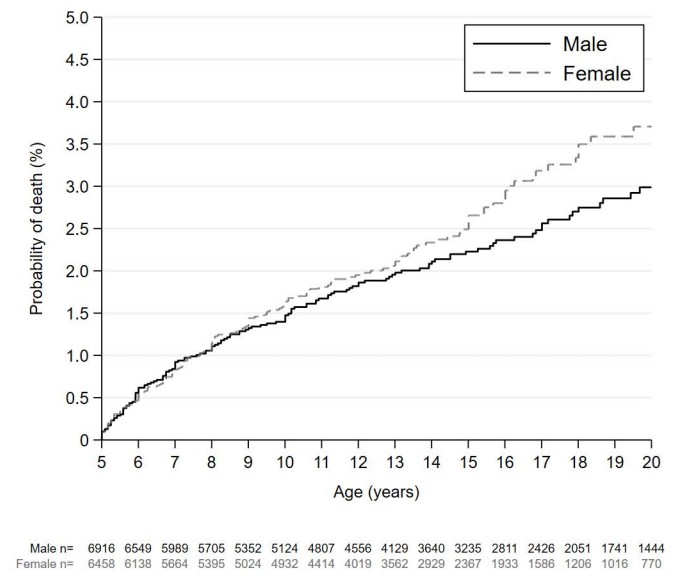

**Fig 1. Probability of death among males and females between ages 5–19 years.**

[CI] 0.98–2.66, p = 0.061 and aHR 1.83 for unimproved and untreated versus improved and treated, 95% CI 1.11–3.02, p = 0.018). Estimates for these risk factors were driven by associations among male 5 to 9-year-olds in bi- and multi-variate models (see S3 Table), in which household infrastructural features—water source and wall materials—emerged as contributing factors. All covariates included in our overall and sex-specific final models had a variance inflation factor <2 indicating multicollinearity was not an issue.

## Factors associated with death in adolescents (ages 10–19 years)

In the five years prior to baseline, a total of 108 adolescent deaths occurred at a rate of 1.90 per 1,000 person years (1.69 and 2.17 per 1,000 person years for males and females, respectively; further disaggregated rates are given in S1 and S2 Tables). Table 2 shows that adolescents living in households in the poorest wealth quintile had a higher risk of death than those living in the wealthiest households (aHR 1.76 for poorest versus wealthiest, 95% CI 0.99–3.14, p = 0.055). Adolescents living in households in which women had received no schooling had a higher risk of death than those living in households in which women had received schooling (aHR 3.52 for no schooling versus any schooling, 95% CI 1.29–9.62, p = 0.014). Maternal education emerged as a strong contributing risk factor among females (see S3 Table), driving the overall association among all adolescents. All covariates included in our overall and sex-specific final models had a variance inflation factor <2 indicating multicollinearity was not an issue.

## Marriage, contraception, and pregnancy in female adolescents of reproductive age (15–19 years)

Of 3,143 females aged 15–19 years in our study population, 543 (17.3%) had information available on marriage, contraception, and pregnancy (Table 3). Of these, 147 (27.1%) were married with a median age at first marriage of 18.0 years (IQR 16.0–18.0). Overall, 10 (3.7%) reproductive age female adolescents were using a method of contraception. Among those who were married,

**Table 2. Age, sex and household characteristics associated with mortality among 5 to 9-year-olds (N = 23,485) and 10 to 19-year-olds (N = 17,910).**

| Characteristic | Categories | Ages 5 to 9 years-old | | | | Categories | Ages 10 to 19 years-old | | | |
|---|---|---|---|---|---|---|---|---|---|---|
| | | Univariate hazard ratio (95%CI) | p | Multivariate hazard ratio (95%CI) | p | | Univariate hazard ratio (95%CI) | p | Multivariate hazard ratio (95%CI) | p |
| **Age and sex** | Male, 5–7 years | 1.00 | | 1.00 | | Male, 10–14 years | 1.00 | | 1.00 | |
| | Male, 8–9 years | 0.57 (0.34, 0.95) | 0.031 | 0.57 (0.34, 0.96) | 0.033 | Male, 15–19 years | 1.19 (0.68, 2.08) | 0.550 | 1.16 (0.67, 2.04) | 0.594 |
| | Female, 5–7 years | 1.00 (0.72, 1.40) | 1.000 | 1.00 (0.72, 1.40) | 0.979 | Female, 10–14 years | 1.07 (0.68, 1.70) | 0.766 | 1.10 (0.70, 1.75) | 0.677 |
| | Female, 8–9 years | 0.85 (0.51, 1.41) | 0.530 | 0.86 (0.52, 1.42) | 0.554 | Female, 15–19 years | 2.27 (1.28, 4.04) | 0.005 | 2.28 (1.29, 4.02) | 0.005 |
| **Ethnicity** | Dogon | 1.00 | | | | Dogon | 1.00 | | | |
| | Fulani | 0.61 (0.21, 1.74) | 0.357 | | | Fulani | 0.79 (0.29, 2.14) | 0.646 | | |
| | Other | 2.41 (1.25, 4.65) | 0.009 | | | Other | 0.43 (0.06, 3.04) | 0.397 | | |
| **Wealth quintile** | Wealthiest | 1.00 | | | | Wealthiest | 1.00 | | 1.00 | |
| | Wealthy | 1.68 (1.03, 2.75) | 0.037 | | | Wealthy | 0.54 (0.23, 1.24) | 0.144 | 0.52 (0.22, 1.19) | 0.123 |
| | Middle | 1.19 (0.71, 2.00) | 0.514 | | | Middle | 0.97 (0.48, 1.95) | 0.936 | 0.92 (0.46, 1.85) | 0.814 |
| | Poor | 1.11 (0.66, 1.85) | 0.701 | | | Poor | 1.37 (0.72, 2.60) | 0.344 | 1.27 (0.66, 2.43) | 0.478 |
| | Poorest | 0.98 (0.58, 1.68) | 0.951 | | | Poorest | 1.79 (1.01, 3.19) | 0.048 | 1.76 (0.99, 3.14) | 0.055 |
| | Unknown | – | | | | Unknown | – | | – | |
| **Decision making contribution of women in household** | Contribute | 1.00 | | | | Contribute | 1.00 | | | |
| | Do not contribute | 1.07 (0.76, 1.51) | 0.678 | | | Do not contribute | 0.98 (0.63, 1.53) | 0.937 | | |
| | Unknown | – | | | | Unknown | – | | | |
| **Highest level of reading ability among women in household** | Can read | 1.00 | | | | Can read | nc | | | |
| | Can partly read | 0.77 (0.19, 3.04) | 0.706 | | | Can partly read | 1.00 | | | |
| | Cannot read | 1.17 (0.52, 2.62) | 0.699 | | | Cannot read | 1.68 (0.42, 6.81) | 0.465 | | |
| | Unknown | – | | | | Unknown | – | | | |
| **Highest level of schooling among women in household** | Schooling | 1.00 | | | | Schooling | 1.00 | | 1.00 | |
| | No schooling | 1.01 (0.60, 1.69) | 0.970 | | | No schooling | 3.29 (1.21, 8.98) | 0.020 | 3.52 (1.29, 9.62) | 0.014 |
| | Unknown | – | | | | Unknown | – | | – | |
| **Polygamy** | Monogamous | 1.00 | | | | Monogamous | 1.00 | | | |
| | Polygamous | 1.15 (0.84, 1.59) | 0.375 | | | Polygamous | 1.31 (0.86, 1.99) | 0.210 | | |
| | Unknown | – | | | | Unknown | – | | | |
| **Domestic violence** | Not tolerated | 1.00 | | | | Not tolerated | 1.00 | | | |
| | Tolerated | 1.42 (0.94, 2.14) | 0.094 | | | Tolerated | 0.95 (0.57, 1.59) | 0.853 | | |
| | Unknown | – | | | | Unknown | – | | | |
| **Water source** | Improved and treated | 1.00 | | 1.00 | | Improved and treated | 1.00 | | | |
| | Improved but untreated | 1.62 (0.98, 2.67) | 0.060 | 1.61 (0.98, 2.66) | 0.061 | Improved but untreated | 1.41 (0.73, 2.74) | 0.307 | | |
| | Unimproved but treated | 1.13 (0.50, 2.55) | 0.766 | 1.13 (0.50, 2.54) | 0.772 | Unimproved but treated | 2.47 (1.15, 5.32) | 0.021 | | |
| | Unimproved and untreated | 1.83 (1.11, 3.02) | 0.017 | 1.83 (1.11, 3.02) | 0.018 | Unimproved and untreated | 1.70 (0.86, 3.35) | 0.126 | | |
| | Unknown | – | | – | | Unknown | – | | | |
| **Sanitation** | Improved | 1.00 | | | | Improved | 1.00 | | | |
| | Unimproved | 1.11 (0.81, 1.53) | 0.501 | | | Unimproved | 0.92 (0.61, 1.39) | 0.704 | | |
| | Unknown | – | | | | Unknown | – | | | |

*(Continued)*

**Table 2.** (Continued)

| Characteristic | Categories | Ages 5 to 9 years-old | | | | Categories | Ages 10 to 19 years-old | | | |
|---|---|---|---|---|---|---|---|---|---|---|
| | | Univariate hazard ratio (95%CI) | p | Multivariate hazard ratio (95%CI) | p | | Univariate hazard ratio (95%CI) | p | Multivariate hazard ratio (95%CI) | p |
| Roofing material | Finished | 1.00 | | | | Finished | 1.00 | | | |
| | Rudimentary | 0.86 (0.47, 1.59) | 0.633 | | | Rudimentary | 1.27 (0.64, 2.50) | 0.496 | | |
| | Natural | 1.02 (0.45, 2.29) | 0.966 | | | Natural | 1.13 (0.42, 3.02) | 0.811 | | |
| | Unknown | – | | | | Unknown | – | | | |
| Wall material | Finished | 1.00 | | | | Finished | 1.00 | | | |
| | Rudimentary | 1.62 (0.97, 2.72) | 0.064 | | | Rudimentary | 1.56 (0.78, 3.15) | 0.212 | | |
| | Natural | 1.34 (0.96, 1.87) | 0.088 | | | Natural | 1.10 (0.71, 1.72) | 0.659 | | |
| | Unknown | – | | | | Unknown | – | | | |
| Electricity | No | 1.00 | | | | No | 1.00 | | | |
| | Yes | 1.02 (0.74, 1.40) | 0.917 | | | Yes | 0.91 (0.59, 1.39) | 0.653 | | |
| Primary cooking fuel | Wood | 1.00 | | | | Wood | 1.00 | | | |
| | Straw | 1.26 (0.82, 1.94) | 0.295 | | | Straw | 1.19 (0.71, 2.00) | 0.518 | | |
| | Animal dung | 0.50 (0.07, 3.60) | 0.495 | | | Animal dung | 1.80 (0.46, 7.10) | 0.402 | | |
| | Other | 0.80 (0.12, 5.41) | 0.815 | | | Other | nc | | | |
| Food shortage in past 30 days | No | 1.00 | | | | No | 1.00 | | | |
| | Yes | 0.79 (0.50, 1.24) | 0.300 | | | Yes | 1.00 (0.58, 1.73) | 0.994 | | |
| Livestock | No | 1.00 | | | | No | 1.00 | | | |
| | Yes | 0.88 (0.48, 1.60) | 0.672 | | | Yes | 1.81 (0.58, 5.68) | 0.308 | | |
| Motorized transport | No | 1.00 | | | | No | 1.00 | | | |
| | Yes | 1.12 (0.82, 1.53) | 0.485 | | | Yes | 0.86 (0.57, 1.28) | 0.450 | | |
| Nearest healthcare center, kilometers | <2 | 1.00 | | | | <2 | 1.00 | | | |
| | 2–4.99 | 1.40 (0.83, 2.39) | 0.211 | | | 2–4.99 | 1.07 (0.56, 2.04) | 0.832 | | |
| | 5–6.99 | 1.26 (0.74, 2.15) | 0.400 | | | 5–6.99 | 0.97 (0.52, 1.83) | 0.931 | | |
| | 7–9.99 | 1.29 (0.75, 2.22) | 0.354 | | | 7–9.99 | 0.86 (0.44, 1.69) | 0.658 | | |
| | ≥10 | 1.30 (0.70, 2.41) | 0.399 | | | ≥10 | 1.45 (0.75, 2.80) | 0.268 | | |

Children and adolescents with missing covariate data were included in analyses but hazard ratios for missing categories are not reported. CI, confidence interval; nc, non-calculable.

**Table 3. Marriage, contraception and pregnancy among females aged 15–19 years.**

| | | Female 15 to 19-year-olds (N = 543) |
|---|---|---|
| **Marriage** | Unmarried | 390 (71.8) |
| | Married | 147 (27.1) |
| | Unknown | 6 (1.1) |
| **Age at first marriage, years** | Median (IQR) | 18.0 (16.0, 18.0) |
| | Mean (SD) | 17.3 (1.4) |
| **Contraceptive use*** | None | 523 (96.3) |
| | Hormonal contraception | 7 (1.3) |
| | Condom | 1 (0.2) |
| | Other | 2 (0.4) |
| | Unknown | 10 (1.8) |
| **Previously pregnant** | Yes | 7 (1.3) |
| **Currently pregnant** | Yes | 6 (1.1) |

All values are n (%N) unless otherwise specified. IQR, interquartile range; SD, standard deviation.

*9 of 10 women using contraception were unmarried.

1 (0.7%) was using a method of contraception. Thirteen (2.4%) reproductive age female adolescents were pregnant at the time of the survey (6, 1.1%) or had previously given birth (7, 1.3%).

## Discussion

Amid the dearth of population health data on 5 to 9-year-olds and adolescents in LMICs, our measurements contribute to filling this gap, highlighting high mortality as children age into adolescence and early adulthood in rural Mali. From our trial's baseline census of all households residing in our study area, we found that, although mortality rates generally declined from early childhood to later childhood and adolescence, 5 to 9-year-old and adolescent mortality remain high. Compared to 2016 estimates of global all-cause mortality rates of 1, 0.7, and 1.2 deaths per 1000 person years for children aged 5–9 years, 10–14 years, and 15–19 years, respectively [7], mortality among 5 to 9-year-olds in Bankass was three times as high, while mortality among adolescents was about twice as high. Five to 9-year-old mortality in our study was elevated compared to the African regional average (3.1 vs. 2.0 deaths per 1000 person years), but adolescent mortality was comparable (1.9 deaths per 1000 person years among 10–19 year-olds in our study vs. 1.5 and 2.6 per 1000 person years among 10–14 and 15–19 year-olds, respectively, in the African Region) [7].

Notably, the risk of mortality was especially acute among female adolescents compared to their male counterparts. Known indicators of poverty such as poor infrastructure and low wealth—which increase mortality by limiting access to appropriate healthcare—contributed overall. Low maternal education within the household was a uniquely contributing factor for females. This difference in mortality by sex reverses the trend observed in our earlier analysis of under-fives in which mortality among males was greater than females [17]. Previous work has shown that biological mechanisms lead to greater mortality in males among under-fives [23,24]. Of available global average trends, including among high-income countries and LMICs, the risk of death among males increases overall and in relation to females during adolescence; only within African and Southeast Asian countries do females experience greater mortality into young adulthood [25], similar to what we observed in rural Mali.

This trend reversal and disparity underscores the importance of the confluence of risk factors, such as early marriage, childbearing, access to reproductive health, and gender-based violence, among many other inter-related dimensions of gender inequities (e.g., literacy, education, economic and financial dependence) that uniquely affect access to healthcare and directly increase mortality risk among adolescent girls and young women in LMICs [26]. Accordingly, we also found that low maternal education was a unique driving risk factor among females. This may be indicative of gender inequalities that disadvantage women intergenerationally. Our data did not permit a more detailed analysis of individual-level risk factors associated with female adolescent mortality, a limitation for understanding the significance of this result. Further, in comparison to Mali nationally, a lower percentage of females 15–19 in Bankass were married (27.1% vs. 39.7%), a surprising finding given the extreme poverty and poor health indicators for the region which are often strongly associated with early marriage. However, our sample of females aged 15–19 years in Bankass was small, as a larger number of older adolescent females than males were reported by their household to migrate out of the study area, which may be for reasons of marriage or employment, and also limits measurement accuracy. More research with a larger, in-depth survey of individuals is needed to fully assess and disentangle these risk factors for both males and females in our study population.

## Limitations

Our findings should be interpreted considering several additional limitations. Mortality was based on household members' recall of deaths in the preceding five years, which may be

subject to recall errors, particularly for deaths occurring early in the five-year period. This may have led to a slight under- or overestimate of mortality rates. Elucidation of individual and household risk factors were further limited by the scope of survey questions, which do not represent a comprehensive assessment, particularly for individual-level factors (e.g., physical violence) that other studies have found to importantly contribute to 5 to 9-year-old and adolescent mortality [7,8]. Finally, we are unable to further ascribe the causes and contributors to mortality, which may require more in-depth verbal autopsy approaches in the absence of vital registration data.

## Conclusion

Limitations notwithstanding, our data is unique among population health surveys in its comprehensive census of a large population in a resource-poor area of the world with one of the highest burdens of child mortality. Our findings emphasize the importance of ensuring policy and program advances in rural Mali and similar parts of sub-Saharan Africa address the needs of 5 to 19-year-olds. Interventions focused on improving household infrastructure and maternal education may be especially impactful. Our results also point to the critical need for further research and health programming to improve survival among 5 to 19-year-olds—in particular, a better accounting and measurement of their health outcomes and associated risk factors. The lack of attention to this gap does little to improve global health equity and must be urgently addressed, beginning with inclusion of these groups in population health research.

## Supporting information

**S1 Table. Rates of mortality by age, sex and household characteristics among 5 to 9-year-olds and 10 to 19-year-olds.**
(DOCX)

**S2 Table. Rates of mortality by age and household characteristics among males and females aged 5–19 years.**
(DOCX)

**S3 Table. Age and household characteristics associated with mortality among males and females aged 5–19 years.**
(DOCX)

## Acknowledgments

We are grateful to all study participants. We would also like to acknowledge Djoumé Diakité, Youssouf Keita, Aminata Konipo, and Seydou Sidibé of Muso for their role in implementing the trial and assuring adherence to protocol. This work was presented in preliminary form at the Health Systems Research conference in Bogota (Oct 31–Nov 4, 2022).

## Author contributions

**Conceptualization:** David C. Boettiger.

**Data curation:** Yacouba Samake, Oumar Tolo, Emily Treleaven, Belco Poudiougou, Caroline Whidden, Ari Johnson, Kassoum Kayentao.

**Formal analysis:** Jenny X. Liu, Emily Treleaven, David C. Boettiger.

**Funding acquisition:** Caroline Whidden, Ari Johnson, Kassoum Kayentao.

**Project administration:** Jenny X. Liu.

**Supervision:** David C. Boettiger.

**Writing – original draft:** Jenny X. Liu.

**Writing – review & editing:** Jenny X. Liu, Yacouba Samake, Oumar Tolo, Emily Treleaven, Belco Poudiougou, Caroline Whidden, Ari Johnson, Kassoum Kayentao, David C. Boettiger.

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
