## [Decision Letter · Decision Letter 0]

22 Jan 2024

PGPH-D-23-01206

Older child and adolescent mortality in rural Mali

Dear Dr. Liu,

Thank you for submitting your manuscript to PLOS Global Public Health. After careful consideration, we feel that it has merit but does not fully meet PLOS Global Public Health’s publication criteria as it currently stands. Therefore, we invite you to submit a revised version of the manuscript that addresses the points raised during the review process.

The article is an effort towards addressing the gaps in research on older children and adolescents in LMICs, specifically in rural Mali. Though the research methods and the results are presented clearly, there is a need for improvement in discussing the potential causes of these observed mortality trends, and also to include suggested detailed policy changes with their implications and suggestions for  future research in this important area of public health. Further, expanding on the limitations of the study is also required.

We look forward to receiving your revised manuscript.

Kind regards,

Sangeeta Sharma

Academic Editor

Journal Requirements:

1. We have noticed that you have a list of Supporting Information legends in your manuscript. However, there are no corresponding files uploaded to the submission. Please upload them as separate files with the item type 'Supporting Information'. 

Additional Editor Comments (if provided):

Reviewers' comments:

Reviewer's Responses to Questions

**Comments to the Author**

1. Does this manuscript meet PLOS Global Public Health’s publication criteria ? Is the manuscript technically sound, and do the data support the conclusions? The manuscript must describe methodologically and ethically rigorous research with conclusions that are appropriately drawn based on the data presented.

Reviewer #1: Partly

Reviewer #2: Yes

2. Has the statistical analysis been performed appropriately and rigorously?

Reviewer #1: Yes

Reviewer #2: Yes

3. Have the authors made all data underlying the findings in their manuscript fully available (please refer to the Data Availability Statement at the start of the manuscript PDF file)?

Reviewer #1: Yes

Reviewer #2: Yes

4. Is the manuscript presented in an intelligible fashion and written in standard English?

Reviewer #1: Yes

Reviewer #2: Yes

5. Review Comments to the Author

Reviewer #1: The introduction of the article effectively sets the context and highlights the research gap concerning older children and adolescents in LMICs. It emphasizes the significance of studying this population given their role in shaping future health and well-being. The introduction provides a comprehensive overview of the existing literature and the specific issues related to this age group in the study area, which is crucial for understanding the context of the research.

The methods section is detailed and well-structured, providing a clear explanation of the data sources, inclusion criteria, variable definitions, and statistical analysis. The authors have justified their approach to conducting the research, including the use of household survey data and the rationale for analyzing older children and adolescents separately. However, there are a few points to consider for clarification and improvement:

1. The authors mention "lifetime birth histories" without explaining what this entails. It would be helpful to provide a brief description or reference.

2. While the article mentions ethical approval, it would be beneficial to provide more details about the ethical considerations, especially when working with sensitive data.

3. The article discusses the use of global positioning technology for geographic coordinates but does not provide information on how this data was collected and utilized in the analysis.

The presentation of results is clear and well-organized. The use of Kaplan-Meier curves to visualize mortality trends is effective. The tables and figures provide a concise summary of the findings. However, there are a few areas that could be improved:

1. The discussion of the reversal in mortality trends between males and females during adolescence is intriguing, but the article does not delve into potential explanations or implications. It would be beneficial to include some speculation or hypotheses regarding this observation.

2. While the results highlight the higher risk of mortality among adolescents in poorer households and those with lower maternal education, the article does not explore the potential mechanisms or pathways through which these factors influence mortality. Adding some discussion on these aspects could enhance the interpretation of the findings.

The discussion section effectively addresses the implications of the research and highlights the need for further studies in this area. However, there is room for improvement in the following areas:

1. The discussion could benefit from a more in-depth exploration of the potential causes of the observed mortality trends, especially the higher risk among female adolescents. Speculation or hypotheses regarding the role of early marriage, gender-based violence, or other factors could add depth to the analysis.

2. The limitations of the study are mentioned but could be expanded upon. For example, the potential sources of bias, such as recall bias in reporting deaths, should be discussed in more detail.

3. The authors should consider including a section on policy implications or recommendations based on their findings - "How can the results inform health programming and policies in LMICs, particularly in rural Mali?".

Reviewer #2: This manuscript is an important contribution to knowledge and understanding of mortality in older children and adolescents. In general, the manuscript is well-written and comprehensible.

However, I suggest that you copy edit the manuscript before resubmitting it. Many sentences and paragraphs are overly long, making it difficult to understand them without re-reading. In addition, there are several instances of unclear phrasing, as I have highlighted in the attachment.

I suggest that you create a separate section for the limitations of the study.

In addition, your conclusion focuses only on the dearth of research on the topic of older children and adolescents. I suggest strengthening your conclusion to include how the findings of this study can be used to inform policy and programs in rural Mali and beyond.

6. PLOS authors have the option to publish the peer review history of their article (what does this mean? ). If published, this will include your full peer review and any attached files.

**Do you want your identity to be public for this peer review?** For information about this choice, including consent withdrawal, please see our Privacy Policy .

Reviewer #1: **Yes: ** Ateeb Ahmad Parray

Reviewer #2: **Yes: ** Laura Hoemeke

While revising your submission, please upload your figure files to the Preflight Analysis and Conversion Engine (PACE) digital diagnostic tool, https://pacev2.apexcovantage.com/ . PACE helps ensure that figures meet PLOS requirements. To use PACE, you must first register as a user. Registration is free. Then, login and navigate to the UPLOAD tab, where you will find detailed instructions on how to use the tool. If you encounter any issues or have any questions when using PACE, please email PLOS at figures@plos.org. Please note that Supporting Information files do not need this step.

---

## [Decision Letter · Decision Letter 1]

22 Jul 2024

PGPH-D-23-01206R1

Older child and adolescent mortality in rural Mali

Dear Dr. Liu,

Thank you for submitting your manuscript to PLOS Global Public Health. After careful consideration, we feel that it has merit but does not fully meet PLOS Global Public Health’s publication criteria as it currently stands. Therefore, we invite you to submit a revised version of the manuscript that addresses the points raised during the review process.

Thank you for addressing the reviewers' concerns. At this time, please can you just address the reviewer's concern below regarding the use of the term "older children", which does not seem appropriate for the 5-9 age group. Your title, for example, could simply be more specific about the age range considered in this study. The reviewer's second point is optional.

We look forward to receiving your revised manuscript.

Kind regards,

Marianne Clemence

Staff Editor

Journal Requirements:

Additional Editor Comments (if provided):

Reviewers' comments:

Reviewer's Responses to Questions

**Comments to the Author**

1. If the authors have adequately addressed your comments raised in a previous round of review and you feel that this manuscript is now acceptable for publication, you may indicate that here to bypass the “Comments to the Author” section, enter your conflict of interest statement in the “Confidential to Editor” section, and submit your "Accept" recommendation.

Reviewer #3: All comments have been addressed

2. Does this manuscript meet PLOS Global Public Health’s publication criteria ? Is the manuscript technically sound, and do the data support the conclusions? The manuscript must describe methodologically and ethically rigorous research with conclusions that are appropriately drawn based on the data presented.

Reviewer #3: Yes

3. Has the statistical analysis been performed appropriately and rigorously?

Reviewer #3: Yes

4. Have the authors made all data underlying the findings in their manuscript fully available (please refer to the Data Availability Statement at the start of the manuscript PDF file)?

Reviewer #3: Yes

5. Is the manuscript presented in an intelligible fashion and written in standard English?

Reviewer #3: Yes

6. Review Comments to the Author

Reviewer #3: Thanks for giving this opportunity to review this interesting study, which focused on the children mortality in rural Mali. I only have TWO concerns before publication.

1. Using of term "Older child": children are defined as individual under 18 years by UNICEF. However, the author uses "older child" to represent those aged 5-9 years, while it may lead to some confusions. I would suggest to use "child" instead of "older child". On the other hand, obviously, the authors have defined the term clearly. So this may be not a problem.

2. Table 3 seems not related to other parts of this study. May consider moving to SM.

7. PLOS authors have the option to publish the peer review history of their article (what does this mean? ). If published, this will include your full peer review and any attached files.

**Do you want your identity to be public for this peer review?** For information about this choice, including consent withdrawal, please see our Privacy Policy .

Reviewer #3: No

While revising your submission, please upload your figure files to the Preflight Analysis and Conversion Engine (PACE) digital diagnostic tool, https://pacev2.apexcovantage.com/ . PACE helps ensure that figures meet PLOS requirements. To use PACE, you must first register as a user. Registration is free. Then, login and navigate to the UPLOAD tab, where you will find detailed instructions on how to use the tool. If you encounter any issues or have any questions when using PACE, please email PLOS at figures@plos.org. Please note that Supporting Information files do not need this step.

---

## [Decision Letter · Decision Letter 2]

31 Oct 2024

PGPH-D-23-01206R2

Mortality among 5 to 19-year-olds in rural Mali

Dear Dr. Liu,

Thank you for submitting your manuscript to PLOS Global Public Health. After careful consideration, we feel that it has merit but does not fully meet PLOS Global Public Health’s publication criteria as it currently stands. Therefore, we invite you to submit a revised version of the manuscript that addresses the points raised during the review process.

We look forward to receiving your revised manuscript.

Kind regards,

Sunday Adedini, PhD

Academic Editor

Journal Requirements:

Additional Editor Comments (if provided):

Reviewers' comments:

Reviewer's Responses to Questions

**Comments to the Author**

1. If the authors have adequately addressed your comments raised in a previous round of review and you feel that this manuscript is now acceptable for publication, you may indicate that here to bypass the “Comments to the Author” section, enter your conflict of interest statement in the “Confidential to Editor” section, and submit your "Accept" recommendation.

Reviewer #4: (No Response)

2. Does this manuscript meet PLOS Global Public Health’s publication criteria ? Is the manuscript technically sound, and do the data support the conclusions? The manuscript must describe methodologically and ethically rigorous research with conclusions that are appropriately drawn based on the data presented.

Reviewer #4: Yes

3. Has the statistical analysis been performed appropriately and rigorously?

Reviewer #4: Yes

4. Have the authors made all data underlying the findings in their manuscript fully available (please refer to the Data Availability Statement at the start of the manuscript PDF file)?

Reviewer #4: No

5. Is the manuscript presented in an intelligible fashion and written in standard English?

Reviewer #4: Yes

6. Review Comments to the Author

Reviewer #4: I enjoyed reviewing this manuscript. The authors used data from randomized controlled trial to estimate mortality rates and assess risk factors of mortality among 5-9 years old and adolescents in rural Mali. The paper is well-written, clear, and certainly worthy of publication. I also saw that the authors have taken great care to address the comments by previous reviewers.

A minor concern that I would like the authors to address:

Variables that are conceptually similar – In standard surveys, ‘wealth quintile’ is typically measured using a principal component analysis of several indicators of household wealth, such as drinking water source, roofing material, flooring material, toilet facilities, electricity, primary cooking fuel, and ownership of livestock. In the current manuscript, a number of independent variables are somewhat similar conceptually i.e., wealth quintile as compared to water source, roofing material, wall material, flooring material, electricity, primary cooking fuel, and livestock. My question is, should water source, roofing material etc. be included in the analytic models since these variables were used to create wealth quintile? In my opinion, it is redundant including variables that represent the same phenomenon, even if they have different names. Including conceptually similar variables in a study is generally not recommended, but if the authors must include them, I think a justification is warranted. To address this concern, the authors can do one or both of the following: (a) clarify whether some of the variables used in creating wealth quintile are also included as independent variables, and maybe provide a justification; (b) examine the association/ correlations among these somewhat similar variables using appropriate statistical tests. If strongly associated, maybe include only the most relevant variable(s).

Overall, this paper is a valuable contribution and I hope to see it published following revisions.

7. PLOS authors have the option to publish the peer review history of their article (what does this mean? ). If published, this will include your full peer review and any attached files.

**Do you want your identity to be public for this peer review?** For information about this choice, including consent withdrawal, please see our Privacy Policy .

Reviewer #4: **Yes: ** Olusola A. Omisakin

While revising your submission, please upload your figure files to the Preflight Analysis and Conversion Engine (PACE) digital diagnostic tool, https://pacev2.apexcovantage.com/ . PACE helps ensure that figures meet PLOS requirements. To use PACE, you must first register as a user. Registration is free. Then, login and navigate to the UPLOAD tab, where you will find detailed instructions on how to use the tool. If you encounter any issues or have any questions when using PACE, please email PLOS at figures@plos.org. Please note that Supporting Information files do not need this step.

---

## [Decision Letter · Decision Letter 3]

26 Dec 2024

Mortality among 5 to 19-year-olds in rural Mali

PGPH-D-23-01206R3

Dear Dr. Liu,

We are pleased to inform you that your manuscript 'Mortality among 5 to 19-year-olds in rural Mali' has been provisionally accepted for publication in PLOS Global Public Health.

Best regards,

Julia Robinson

Executive Editor

Reviewer Comments (if any, and for reference):

Reviewer's Responses to Questions

**Comments to the Author**

1. If the authors have adequately addressed your comments raised in a previous round of review and you feel that this manuscript is now acceptable for publication, you may indicate that here to bypass the “Comments to the Author” section, enter your conflict of interest statement in the “Confidential to Editor” section, and submit your "Accept" recommendation.

Reviewer #4: All comments have been addressed

2. Does this manuscript meet PLOS Global Public Health’s publication criteria ? Is the manuscript technically sound, and do the data support the conclusions? The manuscript must describe methodologically and ethically rigorous research with conclusions that are appropriately drawn based on the data presented.

Reviewer #4: Yes

3. Has the statistical analysis been performed appropriately and rigorously?

Reviewer #4: Yes

4. Have the authors made all data underlying the findings in their manuscript fully available (please refer to the Data Availability Statement at the start of the manuscript PDF file)?

Reviewer #4: No

5. Is the manuscript presented in an intelligible fashion and written in standard English?

Reviewer #4: Yes

6. Review Comments to the Author

Reviewer #4: (No Response)

7. PLOS authors have the option to publish the peer review history of their article (what does this mean? ). If published, this will include your full peer review and any attached files.

**Do you want your identity to be public for this peer review?** For information about this choice, including consent withdrawal, please see our Privacy Policy .

Reviewer #4: **Yes: ** Olusola A. Omisakin
